# Inflammatory Biomarkers and Breast Cancer Risk: A Systematic Review of the Evidence and Future Potential for Intervention Research

**DOI:** 10.3390/ijerph17155445

**Published:** 2020-07-28

**Authors:** Rebecca D. Kehm, Jasmine A. McDonald, Suzanne E. Fenton, Marion Kavanaugh-Lynch, Karling Alice Leung, Katherine E. McKenzie, Jeanne S. Mandelblatt, Mary Beth Terry

**Affiliations:** 1Department of Epidemiology, Mailman School of Public Health, Columbia University, 722 W 168th St, New York, NY 10032, USA; rk2967@cumc.columbia.edu (R.D.K.); jam2319@cumc.columbia.edu (J.A.M.); 2Herbert Irving Comprehensive Cancer Center, Columbia University Medical Center, 1130 St Nicholas Ave, New York, NY 10032, USA; 3National Toxicology Program Laboratory, National Institute of Environmental Health Sciences, Research Triangle Park, 111 TW Alexander Drive, Durham, NC 27709, USA; suzanne.fenton@nih.gov; 4California Breast Cancer Research Program, University of California, 300 Lakeside Drive, Oakland, CA 94612, USA; Marion.Kavanaugh-Lynch@ucop.edu (M.K.-L.); Katherine.McKenzie@ucop.edu (K.E.M.); 5Sapientiae, 1977 Manzanita Dr, Oakland, CA 94611, USA; gustane@gmail.com; 6Department of Oncology, Cancer Prevention and Control Program, Georgetown-Lombardi Comprehensive Cancer Center, Georgetown University, 3800 Reservoir Rd. NW, Washington, DC 20057, USA

**Keywords:** breast cancer risk, blood inflammatory biomarkers, c-reactive protein, intervention research, prospective epidemiologic studies, systematic review

## Abstract

Measuring systemic chronic inflammatory markers in the blood may be one way of understanding the role of inflammation in breast cancer risk, and might provide an intermediate outcome marker in prevention studies. Here, we present the results of a systematic review of prospective epidemiologic studies that examined associations between systemic inflammatory biomarkers measured in blood and breast cancer risk. From 1 January 2014 to 20 April 2020, we identified 18 unique studies (from 16 publications) that examined the association of systemic inflammatory biomarkers measured in blood with breast cancer risk using prospectively collected epidemiologic data. Only one marker, C-reactive protein, was studied extensively (measured in 13 of the 16 publications), and had some evidence of a positive association with breast cancer risk. Evidence associating other inflammatory biomarkers and more comprehensive panels of markers with the development of breast cancer is limited. Future prospective evidence from expanded panels of systemic blood inflammatory biomarkers is needed to establish strong and independent links with breast cancer risk, along with mechanistic studies to understand inflammatory pathways and demonstrate how breast tissue responds to chronic inflammation. This knowledge could ultimately support the development and evaluation of mechanistically driven interventions to reduce inflammation and prevent breast cancer.

## 1. Introduction

The relationship between chronic inflammation and cancer is complex and bidirectional. In this review, we focus on the role of inflammation in the development of breast cancer. The mechanisms through which chronic inflammation might lead to cancer have been detailed elsewhere [1,2] and include DNA damage and genomic, epigenomic, and cellular alterations and interactions [1,2]. Regular use of aspirin and other non-steroidal anti-inflammatory drugs (NSAIDs) has been consistently associated with reduced risk of cancers, including breast cancer [3], and reduced breast cancer risk in high-risk women with *BRCA1* and *BRCA2* mutations [4]. These observations provide indirect evidence of a potential link between inflammation and risk of developing breast cancer.

Chronic inflammation can occur in response to lifestyle and environmental factors that are associated with breast cancer. For example, weight gain in adulthood and being overweight or obese after menopause are associated with both chronic inflammation and breast cancer risk [5,6,7]. There is also some evidence linking other measures related to metabolic disease, including waist-to-hip ratio, insulin resistance, and lipids, with risk of postmenopausal breast cancer [8,9]. Other exposures associated with breast cancer risk, including environmental chemical exposures and selected dietary content and patterns, might also increase chronic inflammation [10], although evidence is limited by the fact that many of these exposures are difficult to measure in large epidemiologic studies. It is also hypothesized that one pathway linking stress to increased cancer risk is through inducing a chronic inflammatory state [11].

Measuring systemic chronic inflammatory markers in the blood may be one way of understanding the role of inflammation in breast cancer risk and might provide intermediate outcome markers in prevention studies. Inflammatory markers, such as C-reactive protein (CRP), interleukin-6 (IL-6), and tumor necrosis factor-α (TNF- α) are shown to markedly increase in response to infection and tissue damage, as well as in active state of disease [12]. Variations within the reference range also predict the onset of health events, such as cardiovascular disease [13] and disability [14,15] in individuals without an obvious inflammatory stimulus [12]. Both cancer risk and plasma levels of inflammatory markers increase with age [16,17,18], but it is unclear whether variations in levels of different circulating inflammatory markers are associated with increased risk of cancer [12]. Rodent studies provide direct evidence that some environmental exposures and obesity increase inflammatory cellular influx to the mammary gland, cytokine production, and morphological changes [19,20]. Here, we summarize the current evidence from prospective epidemiologic studies on the association between systemic inflammatory biomarkers measured in blood and breast cancer risk. We follow up this review with a broader discussion of different opportunities for understanding the link between inflammatory processes and the development of breast cancer, and the potential for chronic inflammation measures to be used as intermediate outcome markers in intervention studies.

## 2. Materials and Methods

We conducted a systematic review of prospective epidemiologic studies investigating the association between circulating inflammatory biomarkers measured in blood and breast cancer risk. We performed a search of the PubMed database to identify studies published from 1 January 2014 to 20 April 2020. We selected these dates because three meta-analyses were published in 2015, which searched the literature up until December 2014 [21,22] or February 2015 [23], and so we wanted to compare findings from these reviews with more recent studies that used contemporary (broad-base) biomarker measurement methods, since these have changed substantially over time [24]. We restricted our search to studies that were conducted in humans, published in the English language, and prospectively collected blood samples prior to breast cancer incidence. We used the following mesh terms in our literature search: “breast cancer” AND “inflammation” OR “inflammatory biomarker” OR “blood biomarker” OR “fibrinogen” OR “C-reactive protein” OR “adiponectin”. Our initial search of the PubMed database returned 1,465 publications for further screening. Two authors independently reviewed titles and abstracts, which led to the identification of 30 publications for full-text review. Fourteen publications were subsequently excluded due to outcome (*n* = 5 assessed breast cancer survival rather than incidence), exposure ascertainment (*n* = 1 inflammatory biomarker measured in urine rather than blood), or study design (*n* = 8 case-control study). Case-control studies were excluded because temporality cannot be clearly established for biomarkers that may be affected by the disease process. The remaining 16 publications were included in our review. We searched the reference lists of the included publications for additional eligible publications, but no additional studies were identified. We extracted data on study population, study design, sample size, exposure assessment, confounding assessment, and relevant effect estimates, and the corresponding 95% confidence intervals (CIs) from the included publications. We did not conduct a quantitative meta-analysis because there was substantial heterogeneity across studies in terms of which systemic blood biomarkers were evaluated.

## 3. Results

We identified 18 unique studies from 16 publications (one publication included results from a nested case-control study, prospective cohort study, and a meta-analysis [21]) that examined the association of systemic inflammatory biomarkers measured in blood with breast cancer risk using prospectively collected epidemiologic data [21,22,23,25,26,27,28,29,30,31,32,33,34,35,36,37]. This included six nested case-control studies [21,25,27,29,30,32], nine prospective cohort studies [21,26,28,31,33,34,35,36,37], and three meta-analyses published in 2015 [21,22,23]. Our review included only two individual studies [30,36] that were in the previous meta-analyses [21,22,23]. The individual studies in this review (excluding meta-analyses) were conducted in eleven unique cohorts in the United States, Europe (France, Sweden, Norway, Denmark, and Italy), and China (refer to Table 1 for further details on study population, sample size, and study design). CRP, a protein produced by the liver in response to systemic inflammation, was measured in 12 of the 15 new studies [21,25,26,28,30,31,32,34,35,36,37], and was the focus of all three meta-analyses [21,22,23]. Eight of the 12 new studies used a high-sensitivity measure of CRP [21,26,28,31,34,35,36], which captures lower levels of circulating CRP compared to a standard CRP test, while the other four studies used a standard CRP test [25,30,32,37]. Two of the three studies that did not consider CRP evaluated various other individual inflammatory blood biomarkers, including pro-inflammatory cytokines (e.g., TNF-α, IL-1β, IL-6), markers of oxidative stress (e.g., ox-LDL), and factors associated with coagulation (e.g., fibrinogen) [29,33]; the third study evaluated a panel of 28 inflammatory-related proteins (10 chemokines, 12 cytokines, and six growth factors) [27].

Table 2 provides a summary of findings for the association of CRP with breast cancer risk from the three meta-analyses and 12 unique studies, subdivided into the eight different cohorts assessed (three studies used data from the Women’s Health Initiative (WHI) but considered different stratification groups [28,32,34], and two studies used data from the Women’s Health Study (WHS) [21,35]). Each meta-analysis included a common core of prospective studies [30,36,38,39,40,41,42,43,44], and all found a statistically significant positive association between circulating CRP levels and breast cancer risk (Table 2). These estimates ranged from 7% per doubling of CRP concentration (relative risk (RR): 1.07, 95% confidence interval (CI): 1.02, 1.12) [23] to 26% comparing the highest to lowest category of CRP concentration (RR 1.26, 95% CI 1.07, 1.49) [21]. The meta-analysis by Chan et al. found that the positive association with circulating CRP concentration was similar when focused only on postmenopausal breast cancer risk (RR 1.06, 95% CI: 1.01, 1.11); numbers were too small to assess premenopausal breast cancer separately [23]. Chan et al. also reported that the positive association between circulating CRP and breast cancer risk was observed in studies that examined reverse causation by excluding cases diagnosed in early years of follow-up [23]. All three meta-analyses found evidence for heterogeneity between studies (I^2^ = 45–47%) [21,22,23]. The level of control for confounders was identified as a possible source of heterogeneity, as studies that did not adjust for hormone replacement therapy (HRT) use, physical activity, or alcohol use reported, on average, stronger associations than studies that adjusted for these factors [23]. Geographic region, menopausal status, CRP markers, and case diagnosis method were identified as other possible sources of heterogeneity [22]. There was also some evidence for publication or small study bias in the meta-analyses (Egger’s test *p* = 0.08–0.17) [22,23].

Three of the 12 new studies found an overall statistically significant positive association between CRP and overall breast cancer risk, including a population-based cohort study of 44,715 women (baseline age range = 48–67 years) in Denmark (highest versus lowest tertile: RR 1.30, 95% CI: 1.07, 1.57) [26], a population-based cohort study of 8,130 women (mean baseline age = 49.8 years) in Norway (highest versus lowest tertile: hazard ratio (HR) 1.53, 95% CI: 1.03, 2.28) [31], and a cohort study of 19,437 women (mean baseline age = 49.2 years) in China (CRP > 3 versus < 1 mg/L concentration: HR 1.74, 95 CI: 1.01, 2.97) [36]. All three of these studies used a high-sensitivity measure of CRP and adjusted for age, BMI, and smoking status [26,31,36]; additional covariates, including alcohol consumption, physical activity, oral contraceptive use, HRT use, diabetes, and marital status were adjusted for in some, but not all, of these estimates. An additional three studies reported point estimates that were not statistically significant, but >1.10 (HR/odds ratio (OR) ranged from 1.13–1.27) [21,30,32]. Eight of the twelve studies considered associations of CRP with breast cancer risk by different stratifying factors, including menopausal status [25,31,35,36,37], body size [30,34,35], and use of HRT [31,32,35]. Findings were inconsistent across the five studies that examined the association of CRP with breast cancer risk stratified by menopausal status [25,31,35,36,37]. Two studies found a statistically significant positive association between CRP and breast cancer risk for postmenopausal women (highest versus lowest tertile: HR 1.87, 95% CI: 1.17, 2.98 [31] and RR 2.42, 95% CI: 1.17, 5.00 [25]), but not for premenopausal women (HR 0.89, 95% CI: 0.37, 2.15 [31] and RR 0.74, 95% CI: 0.40, 1.37 [25], respectively). In contrast, two studies found a statistically significant positive association between CRP and breast cancer risk for premenopausal women (CRP > 3 versus < 1 mg/L concentration and dichotomized at 10 mg/L concentration, respectively: HR 2.76, 95% CI: 1.18, 6.48 [36] and HR 1.18, 95% CI: 1.08, 1.30 [37]), but not for postmenopausal women (HR 1.34, 95% CI: 0.68, 2.64 [36] and HR 1.00, 95% CI: 0.93, 1.07 [37], respectively). One study found no association for either postmenopausal (per 1 standard deviation (SD) increase in concentration: HR 1.02, 95% CI: 0.93, 1.12) or premenopausal (HR 0.96, 95% CI: 0.84, 1.10) women [35]. Three studies were stratified by body mass index (BMI), but findings were again inconsistent [30,34,35]. For example, Nelson et al. found a statistically significant positive association for postmenopausal women with a BMI <25 kg/m^2^ (per 1 SD increase in natural log concentration: HR 1.17, 95% CI: 1.03, 1.33), but not for postmenopausal women with a BMI ≥ 25 kg/m^2^ (HRs ranged from 0.94–1.04 for postmenopausal women in the highest three BMI categories) [34]. In contrast, Dossus et al. found a statistically significant positive association for postmenopausal women with a BMI of ≥25 kg/m^2^ (per natural log increase in concentration: OR 1.52, 95% CI: 1.16, 2.00), but not for postmenopausal women with a BMI < 25 kg/m^2^ (OR 1.02, 95% CI: 0.86, 1.21) [30]. Dossus et al. also found a statistically significant positive association between CRP and breast cancer risk for postmenopausal women with a waist circumference of ≥88 cm (per natural log increase in concentration: OR 1.74, 95% CI: 1.13, 2.66), but not for postmenopausal women with a waist circumference of <88 cm (OR 1.08, 95% CI: 0.93, 1.26) [30]; this was the only study to stratify by waist circumference. Three studies stratified by HRT use [31,32,35], two of which suggest an elevated risk of postmenopausal breast cancer for non-users of HRT with high CRP levels (highest versus lowest tertile and highest versus lowest quartile, respectively: HR 2.08, 95% CI: 1.16, 3.76 [31] and HR 1.63, 95% CI: 0.95, 2.80 [32]), but not for users of HRT (HR 1.32, 95% CI: 0.57, 3.05 [31] and HR 0.90, 95% CI: 0.53, 1.53 [32]). The third study found no association between CRP and breast cancer risk for non-users (per 1 SD increase in concentration: HR 1.02, 95% CI: 0.92, 1.14), past users (HR 0.85, 95% CI: 0.67, 1.08), or current (HR 1.00, 95% CI: 0.90, 1.11) users of HRT, but these associations were not stratified by menopausal status [35].

Table 3 summarizes the findings from studies that examined associations of other, non-CRP, inflammatory blood biomarkers in association with breast cancer risk. This includes studies that assessed blood levels of pro-inflammatory biomarkers, including factor VII antigen activity [33], factor VII concentration [33], fibrinogen [26,33,35], GlycA (circulating N-acetyl methyl groups) [35], haptoglobin [37], hepatocyte growth factor (HGF) [32], interleukin (IL)-1β [29], IL-6 [25,29,32], IL-8 [29], leptin [25,32], leukocyte count [26], lymphocyte count [29], neutrophil count [29], oxidized (OX)-LDL [29], plasminogen activator inhibitor (PAI)-1 [32], resistin [32], soluble intercellular cell adhesion molecule (sICAM)-1 [35], TNF-α [25,29,32], and white blood cell (WBC) count [28,29,31,37], as well as anti-inflammatory biomarkers, including adiponectin [25,32] and albumin [37]. These biomarkers have not been consistently considered across the studies, and each biomarker was only analyzed in one or up to four of the publications included in this review. None of the three meta-analyses included in this review considered systemic blood biomarkers of inflammation other than CRP.

A statistically significant positive association was found between fibrinogen and breast cancer risk in the WHS cohort (highest versus lowest quintile: HR 1.25, 95% CI: 1.03, 1.51) [35], but not in a population-based cohort in Denmark (highest versus lowest tertile: HR 1.05, 95% CI: 0.87, 1.27) [26], nor in the WHI cohort, which included only postmenopausal women (≥324.5 versus <274.5 mg/dL at baseline: HR 0.92, 95% CI: 0.67, 1.26) [33]. When the WHS cohort was stratified by menopausal status, the statistically significant positive association between fibrinogen and breast cancer risk was found for premenopausal women (per 1 SD increase in concentration: HR 1.19, 95% CI: 1.03, 1.38), but not for postmenopausal women (HR 1.07, 95% CI: 0.98, 1.18) [35]. When the WHS cohort was stratified by BMI, a statistically significant positive association between fibrinogen and breast cancer risk was found for women with a BMI < 25 kg/m^2^ (per 1 SD increase in concentration: HR 1.12, 95% CI: 1.01, 1.24), but not for women with a BMI ≥ 25 kg/m^2^ (HR 1.03, 95% CI: 0.94, 1.14) [35].

Statistically significant positive associations with breast cancer risk were found for IL-1β (a pro-inflammatory cytokine) in a Swedish nested case-control study of postmenopausal women (highest category versus none: HR 1.71, 95% CI: 1.05, 2.79) [29], and for leukocyte count in a population-based cohort in Denmark (highest versus lowest tertile: RR 1.33, 95% CI: 1.11, 1.58) [26]. When the WHI cohort was stratified by HRT use, a statistically significant positive association between PAI-1 (a pro-inflammatory adipokine) and breast cancer risk was found for postmenopausal women who did not use HRT (highest versus lowest quartile: HR 1.71, 95% CI: 1.02, 2.89), but not for postmenopausal women who used HRT (HR 1.17, 95% CI: 0.71, 1.93) [32]. In a separate analysis of the WHI cohort, a statistically significant positive association was found between WBC count and in situ breast cancer risk (dichotomized at 10,000 cells/μL: HR 1.65, 95% CI: 1.17, 2.33) [28].

Several studies found a negative association between specific inflammatory biomarkers in blood and breast cancer risk. A nested case-control study in the European Prospective Investigation into Cancer and the nutrition (EPIC)-Varese cohort found a statistically significant negative association between adiponectin (an anti-inflammatory molecule that is involved in the inhibition of IL-6 production, accompanied by induction of the anti-inflammatory cytokines IL-10 and IL-1 receptor antagonists [45]) and breast cancer risk for postmenopausal women (highest versus lowest tertile: RR 0.37, 95% CI: 0.19, 0.72), but not for premenopausal women (RR 1.11, 95% CI: 0.61, 2.03) [25]. This same study found a statistically significant negative association between leptin, which is shown to be involved in pro-inflammatory activities, such as the protection of T lymphocytes from apoptosis and the modulation of T cell proliferation [45], and breast cancer risk for premenopausal women (highest versus lowest tertile: RR 0.43, 95% CI: 0.20, 0.89), but not for postmenopausal women (RR 1.74, 95% CI: 0.83, 3.63) [25]. A nested case-control study in the Malmö Diet and Cancer cohort of postmenopausal women in Sweden found statistically significant negative associations with breast cancer risk for OX-LDL (highest versus lowest tertile: OR 0.63, 95% CI: 0.45, 0.89) and TNF-α (OR 0.65, 95% CI: 0.43, 0.99) [29]. Finally, an analysis of the WHS cohort found an overall statistically significant negative association between sICAM-1, which mediates leukocyte adhesion and trafficking as part of the immune response and vascular inflammation [35], and breast cancer risk (highest versus lowest quintile: HR 0.79, 95% CI: 0.66, 0.94) [35]. When this study was stratified by HRT use, a statistically significant negative association between sCIAM-1 and breast cancer risk was found for current users of HRT (per 1 SD increase in concentration: HR 0.90, 95% CI: 0.83, 0.98), but not for past or non-users of HRT (HR 0.95, 95% CI: 0.78, 1.15 and HR 0.97, 95% CI: 0.89, 1.06, respectively) [35].

Two studies considered composite scores of inflammatory biomarkers [26,27]. Using a population-based cohort in Denmark, Allin et al. created an inflammatory score based on the number of biomarkers with high concentration levels, defined as levels in the third tertile [26]. Biomarkers included high-sensitivity CRP, fibrinogen and whole blood leukocyte count, and scores ranged from zero to three. Women with an inflammatory score of three had a 42% higher risk of breast cancer compared to women with an inflammatory score of zero (RR 1.42, 95% CI: 1.11, 1.80) [26]. However, this association was found to attenuate as follow-up time increased. For example, when follow-up time was terminated after one year, the hazard ratio comparing women with three versus zero elevated biomarkers was 2.01 (95% CI: 1.17, 3.46) [26]. By contrast, the hazard ratio comparing women with three versus zero elevated biomarkers was 1.33 (95% CI: 0.99, 1.79) after four years of follow-up [26]. The second study to use an inflammatory score was a nested case-control study conducted in the EPIC-Italy and the Northern Sweden Health and Disease Study (NSHDS) cohorts. This study constructed an inflammatory score using a panel of 12 cytokines, 10 chemokines, and six growth factors measured in peripheral blood samples [27]. They also used the first principal component (PC1) from a principal component analysis of the 28 inflammatory markers (PC1 explained 32.6% of the total variance) to further evaluate the association of inflammation with breast cancer risk [27]. This study found an overall negative, but not statistically significant, association with breast cancer risk for both the inflammatory score (β −1.72, 95% CI: −3.86, 0.42) and PC1 (β −1.00, 95% CI: −2.12, 0.12) [27]. When the analysis was stratified by time to diagnosis, a statistically significant negative association was found for both the inflammatory score (β −2.88, 95% CI: −5.47, −0.29) and PC1 (β −1.55, 95% CI: −2.92, −0.18) with breast cancer risk for women diagnosed within six years of baseline [27]. No associations were found for women diagnosed with breast cancer after six years from baseline [27].

## 4. Discussion

We systematically reviewed the literature on circulating blood biomarkers of inflammation available since the last meta-analysis published in 2015. Only one marker, CRP, has been studied extensively. Three meta-analyses published in 2015 each found a consistent, positive association between elevated blood levels of CRP and breast cancer risk. However, for the new studies identified in this updated review, only half of the studies (6 of 12) reported point estimates >1.10 [21,26,30,31,32,36], of which only three were statistically significant [26,31,36]. Positive associations were found for subgroups of women stratified by menopausal status, BMI, or HRT use, although findings were inconsistent across studies. There might be several reasons for the inconsistency in findings across more recent studies, including differences in assay methods (e.g., 8 of 12 studies used a high-sensitivity compared with a standard measure of CRP) and differences in covariate adjustment. For example, physical activity and alcohol consumption were not consistently controlled for across studies. Interestingly, none of the studies in this review adjusted for NSAID use, which could be an important confounder. There were also differences across studies regarding sample size and number of breast cancer events (some studies may have been underpowered, which could explain differences between results from meta-analyses and individual studies), median age at baseline, and median length of follow-up time. Length of follow-up is important to consider when interpreting study results, as long-term follow-up is needed to rule out reverse causality. A few studies in this review found stronger associations of CRP with breast cancer risk in the first few years of follow-up, which might be more indicative of consequence rather than causes of underlying cancer [26,27], but the meta-analysis by Chan et al. found that positive associations remained in studies that excluded early years of follow-up [23]. Given these inconsistencies across studies, further research is needed to confirm the positive relationship between blood levels of CRP and breast cancer risk.

CRP is a sensitive and widely used systemic marker of inflammation that, compared with other inflammatory markers, had several advantages as a chronic inflammation marker for past epidemiologic studies, including the availability of reliable assays and temporal stability [22]. Yet, it is important to remember that CRP is a liver-derived indicator of systemic inflammation, and other inflammatory biomarkers in blood may be more specific to breast-related changes. For example, breast cancer cells are shown to respond to higher circulating levels of certain pro-inflammatory cytokines, such as TNF-α, IL-1β, and IL-6 by increasing the expression of P450 aromatase [29,46]. However, there is currently insufficient epidemiologic evidence for other, non-CRP, inflammatory biomarkers or panels (e.g., inflammation scores) of biomarkers. Further prospective evidence, such as from studies using expanded panels of systemic blood inflammatory biomarkers that reflect breast-specific effects, is thus needed to support a strong and independent link between chronic inflammation and breast cancer risk. In addition to a need for prospective evidence using expanded panels of systemic blood inflammatory biomarkers, understanding the causal role of inflammatory processes in the etiology of breast cancer will require other lines of evidence, including (1) epidemiologic studies on medical conditions related to chronic inflammation and breast cancer risk; (2) animal studies of inflammation and mammary tissue morphology; and (3) studies of breast tissue morphology in humans.

### 4.1. Medical Conditions Related to Chronic Inflammation and Breast Cancer Risk

A growing number of studies have examined whether different medical conditions that are related to chronic inflammation are also associated with breast cancer risk by comparing women with these conditions to women without, in terms of the incidence of breast cancer [47]. However, many different conditions, including rheumatoid arthritis, have been inconsistently associated with breast cancer risk and do not support a strong association between chronic inflammatory conditions and breast cancer risk. The inconsistent evidence between medical conditions related to inflammation and breast cancer risk contrasts with other cancers, such as colorectal cancer, which have a much more consistent relationship with several chronic inflammatory [48]. The inconsistent evidence for breast cancer may be due to the timing of when inflammatory conditions were measured with respect to different windows of susceptibility for breast cancer. For example, inflammation might play a different mechanistic role in postmenopausal breast cancer compared with premenopausal breast cancer, especially given that obesity, which is associated with a state of chronic low-grade inflammation, is associated with a higher risk for postmenopausal breast cancer but a lower risk for premenopausal breast cancer [49,50]. The complex relationships between stress, hormone levels, and inflammation [11,51] might also drive differences in younger and older women, but this area of research remains understudied. It is also important to note that confounding might contribute to age-specific findings, as risk for other chronic diseases that drive inflammation increases with age, along with cancer risk, and plasma levels of inflammatory markers are shown to increase with age [16,17,18]. This review was not able to establish the role of inflammation in postmenopausal breast cancer compared with premenopausal breast cancer, as findings were inconsistent across studies for both subgroups and the lack of repeated measures and consistent time between blood measures and age at diagnoses made it challenging to interpret the findings for women across the menopausal transition. Similar to the studies of inflammatory blood biomarkers and breast cancer risk, there is currently a complete absence of information on the effect of inflammation during key windows of susceptibility on breast cancer risk, even though we know that the pregnancy/lactation cycle induces an inflammatory state akin to wound healing and that breasts also change in form and function around the menopausal transition, a time in life when subgroups of women also experience weight gain and increases in other chronic disease risk factors [10]. Further, in a recent study evaluating transcriptomic pathways differentially expressed in breast tissue samples from overweight/obese (OB) vs. normal weight adolescents, analyses identified inflammatory genes (cytokines CSF1 and IL-10, chemokine receptor CCR2) as among the most highly activated upstream regulators in the OB breast [52]. This suggests that there are innate inflammatory responses within the OB breast even in early life, again supporting a need to study windows of susceptibility. The inconsistent evidence for breast cancer may also be due to the available treatments for chronic conditions, which may affect cancer risk. For example, women with diabetes may be taking metformin, which in turn may reduce their breast cancer risk [53]. Nevertheless, extensive medical data sets within large health systems may make feasible the longitudinal studies that are needed, providing definitive evidence regarding whether selected conditions related to chronic inflammation are also associated with breast cancer risk.

### 4.2. Inflammation and Mammary Tissue Morphology in Animal Models

There is a fair amount of evidence in experimental models of breast cancer that obesity and fatty diet are linked to inflammation within the mammary tissue (macrophage-induced crown-like structures or CLS), morphological increases in fibrous tissue, upregulation of aromatase activity (associated with increased number of fat cells expressing the gene) and other biomarkers of inflammation, such as CC-chemokine ligand 2 (CCL2). CCL2 is an inflammatory cytokine that recruits macrophages to sites of injury. Macrophage-associated CLS has been linked to increased proinflammatory mediators (TNF-α, IL-1β, Cox-2) and aromatase expression, and may be an indicator of mammary tumor risk in rodent models [54]. Caloric restriction reversed CLS presence within the mouse mammary gland, normalized aromatase expression, and decreased proinflammatory indicators in one study of obese mice [55]. Overexpression of CCL2 in transgenic mice had an increased number of macrophages, density of stroma and collagen, and genes encoding matrix remodeling enzymes in their mammary tissue compared to non-transgenic controls, as well as an increased susceptibility to the development of carcinogen-induced mammary tumors [56]. These studies together strengthen evidence that inflammation is not only associated with endogenous gene and protein changes that may be evident in the circulation, but also morphological changes that may be reflected in the breast of women.

### 4.3. Studies of Breast Tissue Morphology in Humans

Critical to understanding the role of chronic inflammation and breast cancer risk is the need to appreciate that breast tissue is metabolically active and can change in form and function with metabolic disease and risk factors. Therefore, along with the need for studies that examine systemic inflammation, studies are needed that examine how the breast responds to chronic inflammation. This will require non-invasive ways to measure breast tissue changes. While standard screening methods, including mammography and magnetic resonance imaging (MRI) without background parenchymal enhancement (BPE), do not capture the metabolic activity in breast tissue, there are alternative non-screening methods that can be used to evaluate metabolic changes in breast tissue in association with inflammation. Such non-screening and non-diagnostic methods include thermography, Diffuse Optical Tomography (DOT), and optical spectroscopy (OS) [57,58,59]. All three of these methods measure tissue vasculature, and hence may be a biosensor or biomarker for metabolic changes in the breast tissue, which can also be achieved with BPE on MRI. Early data support a very strong (~10-fold) [60], and independent, increase in breast cancer risk from BPE separate from breast density, suggesting that measures that map to changes in metabolic activity in the breast may also be important in predicting risk.

These alternative non-invasive methods (thermography, DOT, and OS) do not expose the breast tissue to radiation and can be used to assess changes over short durations of time. Therefore, if validated in different settings, such methods could be used as an outcome for intervention studies, as they represent a non-invasive method to measure vascular changes in the breast tissue. A key advantage of combining these non-invasive measures of breast tissue changes with blood biomarkers is that, in combination, they may be a powerful biosensor and/or intermediate outcome for intervention studies. In addition to being advantageous for intervention studies, these non-invasive measures of breast tissue changes could also be used in etiologic studies to explore the role of chronic inflammation in breast tissue composition and metabolic function.

## 5. Conclusions

The development of a validated panel of chronic inflammatory markers that can be utilized as an intermediate outcome, with or without non-invasive breast tissue measurements, would be useful in intervention studies. Moreover, the use of validated non-invasive biomarkers that are predictive of breast cancer would enhance breast cancer risk reduction and risk stratification. This would be similar to how other chronic diseases are monitored through blood tests combined with intermediate measures (e.g., lipids or blood pressure for heart disease risk). However, this review found that only CRP has been extensively studied in association with breast cancer risk. Few epidemiologic studies have evaluated other inflammatory biomarkers, individually or in panels, that may be more specific to breast-related changes. Future prospective studies utilizing expanded panels of systemic blood inflammatory biomarkers, along with serially collected measurements, are thus needed to strengthen the evidence on inflammation with breast cancer risk. In addition to building the epidemiologic evidence, it will also be important to conduct mechanistic studies to understand inflammatory pathways and demonstrate how breast tissue responds to chronic inflammation, such as through altered metabolic activity. This knowledge could ultimately support the development and evaluation of mechanistically driven interventions to reduce inflammation and prevent breast cancer.

## Figures and Tables

**Table 1 ijerph-17-05445-t001:** Studies of inflammatory biomarkers measured in blood and breast cancer risk, published January 2014 to April 2020 in PubMed.

Unique Study #	Citation #	Author, Year	PMID	Population, Sample Size	Study Design	Biomarkers
1	25	Agnoli, 2017	28983080	351 cases and 351 controls; ages 35–69 years at baseline (1992–1997); 14.9 years of follow-up; Italy	Nested case-control in EPIC-Varese cohort	CRP, TNF-α, IL-6, leptin, adiponectin
2	26	Allin, 2016	27194008	822 cases (*n* = 44,715); ages 48–67 at baseline (2003–2012); median follow-up of 4.8 years; Denmark	Population-based prospective cohort	hsCRP, fibrinogen, leukocyte count, inflammatory score
3	27	Berger, 2018	30018397	167 cases and 249 controls; mean age of 52.8 years at baseline (1990–2008); 15.5 years of follow-up; Italy and Sweden	Nested case-control in EPIC-Italy and NSHDS	Inflammatory score
4	28	Busch, 2018	29614476	394/4,328 invasive and 100/1049 in situ cases (*n* = 14,375/130,844) for CRP/WBC count; 50–79 years at baseline (1993–1998); 18.6 years of follow-up; USA	Prospective cohort study in the Women’s Health Initiative	hsCRP, WBC count
5	23	Chan, 2015	26224798	12 prospective studies involving a total of 3,522 cases (*n* = 69,610); published 2005–2015	Meta-analysis	CRP
6	29	Dias, 2016	27391324	446 cases and 885 controls; ages 55–74 years at baseline (1991–1996); followed through December, 2010; Sweden	Nested case-control in Malmö Diet and Cancer cohort	Ox-LDL, IL-1β, IL-6, IL-8, TNF-α, WBC count, lymphocyte count, neutrophil count
7	30	Dossus, 2014	24504436	549 cases and 1,040 controls; mean age of 57.7/57.4 years (cases/controls) at baseline (1995–1999); followed through July, 2005; France	Nested case-control in French E3N cohort	CRP
8	31	Frydenberg, 2016	26740213	192 cases (*n* = 8,130); mean age of 49.8 years at baseline (1994–2008); 14.6 years of follow-up; Norway	EBBA-Life sub-study in the Tromsø population-based prospective cohort	hsCRP and WBC count
9	32	Gunter, 2015	26185195	875 cases and 839 controls; ages 57–69 years at baseline (1993–1998); followed through June, 2004; USA	Nested case-control in Women’s Health Initiative cohort	CRP, leptin, adiponectin, resistin, IL-6, TNF-α, PAI-1, HGF
10	22	Guo, 2015	26001129	13 prospective studies involving a total of 4,724 cases (*n* = 403,836); published 2005–2014	Meta-analysis	CRP
11	33	Kabat, 2016	26317383	275 cases (*n* = 5,287); 50–79 years at baseline (1993–1998); median follow-up of 11.4 years; USA	Prospective cohort study in the Women’s Health Initiative	Fibrinogen, factor VII antigen activity, factor VII concentration
12	34	Nelson, 2017	28292922	1114 cases (*n* = 17,841); 50–79 years at baseline (1993–1998); mean follow-up of 13.6 years; USA	Prospective cohort study in the Women’s Health Initiative	hsCRP
13	35	Tobias, 2018	28641369	1497 cases (*n* = 27,071); mean age of 54.5 years at baseline (1992–1995); median follow-up of 19 years; USA	Prospective cohort study in the Women’s Health Study	hsCRP, fibrinogen, GlycA, sICAM-1
14	21	Wang, 2015	25994740	943 cases and 1,221 controls; ages 43–69 years at baseline (1989–1990); followed through June, 1998; USA	Nested case-control in the Nurses’ Health Study	hsCRP
15	21	Wang, 2015	25994740	1,919 cases (*n* = 27,900); mean age of 55.6/54.6 years (cases/non-cases) at baseline (1992–1995); median follow-up of 19 years; USA	Prospective cohort study in the Women’s Health Study	hsCRP
16	21	Wang, 2015	25994740	11 prospective studies involving a total of 5,371 cases (*n* = 73,525); published 2006–2015	Meta-analysis	CRP
17	36	Wang, 2015	25490990	87 cases (*n* = 19,437); mean age of 49.2 years at baseline (2006–2007); followed through December, 2011; China	Prospective cohort study in the Chinese Kailuan Female Cohort	hsCRP
18	37	Wulaningsih, 2015	26130675	6606 cases (*n* = 155,179); mean age of 50.3/46.3 years (cases/non-cases) at baseline (1985–1996); mean follow-up of 18.3 years; Sweden	Prospective cohort study in the Apolipoprotein Mortality Risk Study	CRP, WBC count, albumin, haptoglobin

Notes: EPIC = European Prospective Investigation into Cancer and nutrition; NSHDS = Northern Sweden Health and Disease Study. Unique study # distinguishes the 18 different studies from the 16 publications identified in this review and can be used to cross-reference with Table 1. Citation # corresponds to the reference number of each publication in this review and can be cross-referenced with the Reference list at the end of the paper.

**Table 2 ijerph-17-05445-t002:** Association of C-reactive protein (CRP) measured in blood with breast cancer risk from studies published January 2014 to April 2020 in PubMed.

Unique Study # [Citation #]	Author, Year	Biomarker	Analytic Sample	Cases	Estimate	Units of Comparison	Covariates
***Meta-analyses***							
5 [23]	Chan, 2015	CRP	all women in study	3522	**RR 1.07, 95% CI: 1.02, 1.12**	per doubling of concentration	varied by study
		CRP	postmenopausal women	2516	**RR 1.06, 95% CI: 1.01, 1.11**		
10 [22]	Guo, 2015	CRP	all women in study	4724	**OR 1.14, 95% CI: 1.04, 1.25**	per natural log increase in concentration	varied by study
16 [21]	Wang, 2015	CRP	all women in study	5371	**RR 1.26, 95% CI: 1.07, 1.49**	highest vs. lowest category	varied by study
***Women’s Health Initiative***							
4 [28]	Busch, 2018	hsCRP	postmenopausal women	394	HR 1.03, 95% CI: 0.83, 1.27	dichotomized at 3 mg/L	age, race/ethnicity, cohort enrollment, age at menarche, age at menopause, HRT use, breastfeeding, BMI, smoking status, caregiving, negative life events, physical activity, sleep quality
		hsCRP	postmenopausal, in situ cancer	100	HR 1.02, 95% CI: 0.67, 1.55		
9 [32]	Gunter, 2015	CRP	postmenopausal women	875	HR 1.24, 95% CI: 0.86, 1.80	highest vs. lowest quartile	age, BMI, ethnicity, alcohol, family history of BC, parity, years of menstruation, age at first birth, HRT use, endogenous estradiol levels, history of BBD, physical activity
		CRP	postmenopausal, HRT non-users	412	HR 1.63, 95% CI: 0.95, 2.80		
		CRP	postmenopausal, HRT users	463	HR 0.90, 95% CI: 0.53, 1.53		
12 [34]	Nelson, 2017	hsCRP	postmenopausal women	1114	HR 1.05, 95% CI: 0.98, 1.12	per 1 SD increase in natural log concentration	BMI, race/ethnicity, diabetes, hypertension, smoking, HRT use
		hsCRP	postmenopausal, BMI < 25 kg/m^2^	na	**HR 1.17, 95% CI: 1.03, 1.33**		
		hsCRP	postmenopausal, BMI 25–30 kg/m^2^	na	HR 1.04, 95% CI: 0.93, 1.16		
		hsCRP	postmenopausal, BMI 30–35 kg/m^2^	na	HR 0.94, 95% CI: 0.82, 1.08		
		hsCRP	postmenopausal, BMI >35 kg/m^2^	na	HR 0.97, 95% CI: 0.81, 1.16		
***Women’s Health Study***							
13 [35]	Tobias, 2018	hsCRP	all women in study	1497	HR 0.84, 95% CI: 0.69, 1.04	highest vs. lowest quintile	age, BMI, treatment allocation, family history of BC, history of BBD, race/ethnicity, menopausal status, HRT use, age at menarche, parity, age at first birth, OC use, mammography screening, Alternative Healthy Eating Index 2010 score, physical activity, alcohol, smoking, other measured inflammatory biomarkers
		hsCRP	postmenopausal women	859	HR 1.02, 95% CI: 0.93, 1.12	per 1 SD increase in concentration	
		hsCRP	premenopausal women	393	HR 0.96, 95% CI: 0.84, 1.10		
		hsCRP	HRT non-users	682	HR 1.02, 95% CI: 0.92, 1.14		
		hsCRP	HRT past users	134	HR 0.85, 95% CI: 0.67, 1.08		
		hsCRP	HRT current users	679	HR 1.00, 95% CI: 0.90, 1.11		
		hsCRP	women with BMI < 25 kg/m^2^	759	HR 1.02, 95% CI: 0.93, 1.12		
		hsCRP	women with BMI ≥ 25 kg/m^2^	727	HR 0.95, 95% CI: 0.86, 1.06		
15 [21]	Wang, 2015	hsCRP	all women in study	1919	HR 0.89, 95% CI: 0.76, 1.06	highest vs. lowest quintile	BMI, family history of BC, history of BBD, age at menarche, parity, age at first birth, alcohol, smoking, physical activity
***Nurses’ Health Study***							
14 [21]	Wang, 2015	hsCRP	all women in study	943	RR 1.27, 95% CI: 0.93, 1.73	highest vs. lowest quintile	BMI, family history of BC, history of BBD, age at menarche, parity, age at first birth, alcohol, smoking, physical activity
***Women’s Health Study and Nurses’ Health Study***							
14-15 [21]	Wang, 2015	hsCRP	all women in study	2862	RR 1.04, 95% CI: 0.74, 1.46	highest vs. lowest quintile	age, BMI, treatment allocation, menopausal status, HRT use, family history of BC, history of BBD, age at menarche, parity, age at first birth, alcohol, smoking, physical activity
***Population-based cohort in Denmark***							
2 [26]	Allin, 2016	hsCRP	all women in study	822	**RR 1.30, 95% CI: 1.07, 1.57**	highest vs. lowest tertile	age, BMI, physical activity, smoking, alcohol, OC use, HRT use
***Population-based cohort in Norway***							
8 [31]	Frydenberg, 2016	hsCRP	all women in study	192	**HR 1.53, 95% CI: 1.03, 2.28**	highest vs. lowest tertile	age, BMI, number of children, smoking
		hsCRP	postmenopausal women	149	**HR 1.87, 95% CI: 1.17, 2.98**		
		hsCRP	premenopausal women	43	HR 0.89, 95% CI: 0.37, 2.15		
		hsCRP hsCRP	HRT non-users HRT users	130 44	HR 1.69, 95% CI: 0.99, 2.78 HR 0.91, 95% CI: 0.42, 1.99		
		hsCRP	postmenopausal HRT non-users	99	**HR 2.08, 95% CI: 1.16, 3.76**		
		hsCRP	postmenopausal HRT users	37	HR 1.32, 95% CI: 0.57, 3.05		
***Chinese Kailuan Female Cohort***							
17 [36]	Wang, 2015	hsCRP	all women in study	87	**HR 1.74, 95% CI: 1.01, 2.97**	>3 vs. <1 mg/L	age, BMI, smoking, alcohol, diabetes, physical activity, marital status
		hsCRP	postmenopausal women	57	HR 1.34, 95% CI: 0.68, 2.64		
		hsCRP	premenopausal women	30	**HR 2.76, 95% CI: 1.18, 6.48**		
***Apolipoprotein Mortality Risk Study***							
18 [37]	Wulaningsih, 2015	CRP	all women in study	6606	HR 0.99, 95% CI: 0.92, 1.06	dichotomized at 10 mg/L	age, SES
			postmenopausal women	5623	HR 1.00, 95% CI: 0.93, 1.07		
			premenopausal women	3379	**HR 1.18, 95% CI: 1.08, 1.30**		
***Nested case-control in EPIC-Varese***							
1 [25]	Agnoli, 2017	CRP	all women in study	351	RR 1.15, 95% CI: 0.75, 1.76	highest vs. lowest tertile	age, BMI, family history of BC, age at menarche, parity, OC use, smoking education, alcohol
		CRP	postmenopausal women	167	**RR 2.42, 95% CI: 1.17, 5.00**		
		CRP	premenopausal women	180	RR 0.74, 95% CI: 0.40, 1.37		
***Nested case-control in French E3N***							
7 [30]	Dossus, 2014	CRP	postmenopausal women	549	OR 1.13, 95% CI: 0.98, 1.29	per natural log increase in concentration	age, menopausal status, year of blood collection, study center, age at menopause
		CRP	postmenopausal, BMI < 25 kg/m^2^	394	OR 1.02, 95% CI: 0.86, 1.21		
		CRP	postmenopausal, BMI ≥ 25 kg/m^2^	156	**OR 1.52, 95% CI: 1.16, 2.00**		
		CRP	postmenopausal, WC < 88 cm	482	OR 1.08, 95% CI: 0.93, 1.26		
		CRP	postmenopausal, WC ≥ 88 cm	67	**OR 1.74, 95% CI: 1.13, 2.66**		
		CRP	postmenopausal, HC < 97 cm	238	OR 1.14, 95% CI: 0.92, 1.42		
		CRP	postmenopausal, HC ≥ 97 cm	311	OR 1.13, 95% CI: 0.94, 1.37		
		CRP	postmenopausal, WHR < 0.80	383	OR 1.06, 95% CI: 0.89, 1.26		
		CRP	postmenopausal, WHR ≥ 0.80	166	OR 1.28, 95% CI: 0.99, 1.65		

Notes: BBD = benign breast disease; BC = breast cancer; BMI = body mass index; CRP = C-reactive protein; HC = hip circumference; HRT = hormone replacement therapy; hsCRP = high-sensitivity CRP; na = not available; OC = oral contraceptive; SES = socioeconomic status; WC = waist circumference; WHR = waist-to-hip ratio. Unique study # distinguishes the 18 different studies from the 16 publications identified in this review and can be used to cross-reference with Table 1. Citation # corresponds to the reference number of each publication in this review and can be cross-referenced with the Reference list at the end of the paper.

**Table 3 ijerph-17-05445-t003:** Associations of other, non-C-reactive protein (CRP), inflammatory biomarkers measured in blood with breast cancer risk from studies published January 2014 to April 2020 in PubMed.

Unique Study # [Citation #]	Author, Year	Study	Analytic Sample	Cases	Units of Comparison	Estimate	Covariates
***Adiponectin***							
1 [25]	Agnoli, 2017	EPIC-Varese	all women in study	351	highest vs. lowest tertile	RR 0.73, 95% CI: 0.48, 1.11	age, BMI, family history of BC, age at menarche, parity, OC use, smoking, alcohol, education
			postmenopausal women	167		**RR 0.37, 95% CI: 0.19, 0.72**	
			premenopausal women	180		RR 1.11, 95% CI: 0.61, 2.03	

9 [32]	Gunter, 2015	WHI	postmenopausal women	875	highest vs. lowest quartile	HR 0.76, 95% CI: 0.55, 1.06	age, BMI, ethnicity, alcohol, family history of BC, parity, years of menstruation, age at first birth, HRT use, endogenous estradiol levels, history of BBD, physical activity
***Albumin***							
18 [37]	Wulaningsih, 2015	AMRS cohort	all women in study	6606	dichotomized at 40 g/L	HR 0.97, 95% CI: 0.91, 1.05	age, SES
			postmenopausal women	5623		HR 0.95, 95% CI: 0.88, 1.03	
			premenopausal women	3379		HR 0.92, 95% CI: 0.83, 1.02	
***Factor VII antigen activity***							
11 [33]	Kabat, 2016	WHI	postmenopausal women	275	baseline ≥ 135.5 vs. <110.5 mg/dL	HR 1.12, 95% CI: 0.83, 1.52	age, BMI, education, ethnicity, treatment allocation
***Factor VII concentration***							
11 [33]	Kabat, 2016	WHI	postmenopausal women	275	baseline ≥ 135.5 vs. <110.5 mg/dl	HR 1.02, 95% CI: 0.75, 1.38	age, BMI, education, ethnicity, treatment allocation
***Fibrinogen***							
2 [26]	Allin, 2016	Danish cohort	all women in study	822	highest vs. lowest tertile	RR 1.05, 95% CI: 0.87, 1.27	age, BMI, physical activity, smoking, alcohol, OC use, HRT use
11 [33]	Kabat, 2016	WHI	postmenopausal women, baseline measure	275	≥ 324.5 vs. <274.5 mg/dL	HR 0.92, 95% CI: 0.67, 1.26	age, BMI, education, ethnicity, treatment allocation
			postmenopausal women, average measure	260	average ≥ 316.6 vs. <273.1 mg/dL	HR 0.86, 95% CI: 0.63, 1.18	
			postmenopausal women, 1-3 years measure	108		HR 0.80, 95% CI: 1.47, 1.34 *	
			postmenopausal women, 2-4 years measure	100		HR 0.94, 95% CI: 0.56, 1.60	
			postmenopausal women, 3-5 years measure	98		HR 1.14, 95% CI: 0.67, 1.95	
13 [35]	Tobias, 2018	WHS	all women in study	1497	highest vs. lowest quintile	**HR 1.25, 95% CI: 1.03, 1.51**	age, BMI, treatment allocation, family history of BC, history of BBD, race/ethnicity, menopausal status, HRT use, age at menarche, parity, age at first birth, OC use, mammography screening, Alternative Healthy Eating Index 2010 score, physical activity, alcohol, smoking, other measured inflammatory biomarkers
			postmenopausal women	859	per 1 SD increase in concentration	HR 1.07, 95% CI: 0.98, 1.18	
			premenopausal women	393		**HR 1.19, 95% CI: 1.03, 1.38**	
			HRT non-users	682		HR 1.07, 95% CI: 0.96, 1.20	
			HRT past users	134		**HR 1.25, 95% CI: 0.97, 1.60**	
			HRT current users	679		HR 1.05, 95% CI: 0.94, 1.16	
			women with BMI < 25 kg/m^2^	759		HR 1.12, 95% CI: 1.01, 1.24	
			women with BMI ≥ 25 kg/m^2^	727		HR 1.03, 95% CI: 0.94, 1.14	
***GlycA (circulating N-acetyl methyl groups)***							
13 [35]	Tobias, 2018	WHS	all women in study	1497	highest vs. lowest quintile	HR 0.96, 95% CI: 0.79, 1.17	age, BMI, treatment allocation, family history of BC, history of BBD, race/ethnicity, menopausal status, HRT use, age at menarche, parity, age at first birth, OC use, mammography screening, Alternative Healthy Eating Index 2010 score, physical activity, alcohol, smoking, other measured inflammatory biomarkers
			postmenopausal women	859	per 1 SD increase in concentration	HR 0.95, 95% CI: 0.87, 1.03	
			premenopausal women	393		HR 0.97, 95% CI: 0.85, 1.10	
			HRT non-users	682		HR 0.97, 95% CI: 0.88, 1.07	
			HRT past users	134		HR 0.87, 95% CI: 0.70, 1.07	
			HRT current users	679		HR 1.02, 95% CI: 0.92, 1.13	
			women with BMI < 25 kg/m^2^	759		HR 1.00, 95% CI: 0.91, 1.10	
			women with BMI ≥ 25 kg/m^2^	727		HR 0.96, 95% CI: 0.87, 1.06	
***Haptoglobin***							
18 [37]	Wulaningsih, 2015	AMRS cohort	all women in study	4764	dichotomized at 1.4 g/L	HR 1.09, 95% CI: 1.00, 1.18	age, SES
			postmenopausal women	4113		HR 1.09, 95% CI: 1.00, 1.19	
			premenopausal women	2514		HR 0.94, 95% CI: 0.83, 1.07	
***Hepatocyte growth factor***							
9 [32]	Gunter, 2015	WHS	postmenopausal women	874	highest vs. lowest quartile	HR 1.20, 95% CI: 0.87, 1.65	age, BMI, ethnicity, alcohol, family history of BC, parity, years of menstruation, age at first birth, HRT use, endogenous estradiol levels, history of BBD, physical activity
***Inflammatory Score***							
2 [26]	Allin, 2016	Danish cohort	all women in study	822	3 vs. 0 high inflammatory markers	**RR 1.42, 95% CI: 1.11, 1.80**	age, BMI, physical activity, smoking, alcohol, OC use, HRT use

3 [27]	Berger, 2018	EPIC-Italy and NSHDS	all women in study	167	score difference in cases and controls	β −1.72, 95% CI: −3.86, 0.42	age, study center, BMI, smoking, alcohol, physical activity, education, menopausal status, contraceptive use, age at menarche, HRT use, parity
			time to diagnosis ≤ 6 years	49		**β −2.88, 95% CI: −5.47, −0.29**	
			time to diagnosis > 6 years	41		β −0.06, 95% CI: −2.86, 2.74	
			all women in study	167	PC1 difference in cases and controls	β −1.00, 95% CI: −2.12, 0.12	
			time to diagnosis ≤ 6 years	49		**β −1.55, 95% CI: −2.92, −0.18**	
			time to diagnosis > 6 years	41		β −0.09, 95% CI: −1.56, 1.38	
***Interleukin-1β***							
6 [29]	Dias, 2016	MDC cohort	postmenopausal women	446	highest category vs. none	**OR 1.71, 95% CI: 1.05, 2.79**	age, week of blood sampling, BMI, WHR, HRT use, parity, smoking, alcohol, physical activity, education
***Interleukin-6***							
1 [25]	Agnoli, 2017	EPIC-Varese	all women in study	351	highest vs. lowest tertile	RR 1.58, 95% CI: 0.89, 2.82	age, BMI, family history of BC, age at menarche, parity, OC use, smoking, alcohol, education
			postmenopausal women	167		RR 1.53, 95% CI: 0.59, 3.96	
			premenopausal women	180		RR 1.89, 95% CI: 0.83, 4.28	
9 [32]	Gunter, 2015	WHI	postmenopausal women	856	highest vs. lowest quartile	HR 1.20, 95% CI: 0.85, 1.69	age, BMI, ethnicity, alcohol, family history of BC, parity, years of menstruation, age at first birth, HRT use, endogenous estradiol levels, history of BBD, physical activity
6 [29]	Dias, 2016	MDC cohort	postmenopausal women	446	highest vs. lowest tertile	OR 0.80, 95% CI: 0.56, 1.15	age, week of blood sampling, BMI, WHR, HRT use, parity, smoking, alcohol, physical activity, education
***Interleukin-8***							
6 [29]	Dias, 2016	MDC cohort	postmenopausal women	446	highest vs. lowest tertile	OR 1.09, 95% CI: 0.71, 1.66	age, week of blood sampling, BMI, WHR, HRT use, parity, smoking, alcohol, physical activity, education
***Leptin***							
1 [25]	Agnoli, 2017	EPIC-Varese	all women in study	351	highest vs. lowest tertile	RR 0.83, 95% CI: 0.51, 1.37	age, BMI, family history of BC, age at menarche, parity, OC use, smoking, alcohol, education
			postmenopausal women	167		RR 1.74, 95% CI: 0.83, 3.63	
			premenopausal women	180		**RR 0.43, 95% CI: 0.20, 0.89**	
9 [32]	Gunter, 2015	WHI	postmenopausal women	874	highest vs. lowest quartile	HR 1.39, 95% CI: 0.93, 2.09	age, BMI, ethnicity, alcohol, family history of BC, parity, years of menstruation, age at first birth, HRT use, endogenous estradiol levels, history of BBD, physical activity
***Leukocyte count***							
2 [26]	Allin, 2016	Danish cohort	all women in study	822	highest vs. lowest tertile	**RR 1.33, 95% CI: 1.11, 1.58**	age, BMI, physical activity, smoking, alcohol, OC use, HRT use
***Lymphocyte**count***							
6 [29]	Dias, 2016	MDC cohort	postmenopausal women	446	highest vs. lowest tertile	OR 0.94, 95% CI: 0.68, 1.28	age, week of blood sampling, BMI, WHR, HRT use, parity, smoking, alcohol, physical activity, education
***Neutrophil count***							
6 [29]	Dias, 2016	MDC cohort	postmenopausal women	446	highest vs. lowest tertile	OR 1.04, 95% CI: 0.74, 1.46	age, week of blood sampling, BMI, WHR, HRT use, parity, smoking, alcohol, physical activity, education
***Oxidized-LDL***							
6 [29]	Dias, 2016	MDC cohort	postmenopausal women	446	highest vs. lowest tertile	**OR 0.63, 95% CI: 0.45, 0.89**	age, week of blood sampling, BMI, WHR, HRT use, parity, smoking, alcohol, physical activity, education
***Plasminogen activator inhibitor-1***							
9 [32]	Gunter, 2015	WHI	postmenopausal women	858	highest vs. lowest quartile	HR 1.33, 95% CI: 0.96, 1.86	age, BMI, ethnicity, alcohol, family history of BC, parity, years of menstruation, age at first birth, HRT use, endogenous estradiol levels, history of BBD, physical activity
			postmenopausal, HRT non-users	403		**HR 1.71, 95% CI: 1.02, 2.89**	
			postmenopausal, HRT users	455		HR 1.17, 95% CI: 0.71, 1.93	
***Resistin***							
9 [32]	Gunter, 2015	WHI	postmenopausal women	875	highest vs. lowest quartile	HR 0.93, 95% CI: 0.68, 1.27	age, BMI, ethnicity, alcohol, family history of BC, parity, years of menstruation, age at first birth, HRT use, endogenous estradiol levels, history of BBD, physical activity
***Soluble intercellular cell adhesion molecule-1***							
13 [35]	Tobias, 2018	WHS	all women in study	1497	highest vs. lowest quintile	**HR 0.79, 95% CI: 0.66, 0.94**	age, BMI, treatment allocation, family history of BC, history of BBD, race/ethnicity, menopausal status, HRT use, age at menarche, parity, age at first birth, OC use, mammography screening, Alternative Healthy Eating Index 2010 score, physical activity, alcohol, smoking, other measured inflammatory biomarkers
			postmenopausal women	859	per 1 SD increase in concentration	HR 0.95, 95% CI: 0.88, 1.02	
			premenopausal women	393		HR 0.96, 95% CI: 0.86, 1.08	
			HRT non-users	682		HR 0.97, 95% CI: 0.89, 1.06	
			HRT past users	134		HR 0.95, 95% CI: 0.78, 1.15	
			HRT current users	679		**HR 0.90, 95% CI: 0.83, 0.98**	
			women with BMI < 25 kg/m^2^	759		HR 0.93, 95% CI: 0.86, 1.01	
			women with BMI ≥ 25 kg/m^2^	727		HR 0.94, 95% CI: 0.86,1.01	
***Tumor necrosis factor-α***							
1 [25]	Agnoli, 2017	EPIC-Varese	all women in study	351	highest vs. lowest tertile	RR 1.36, 95% CI: 0.79, 2.34	age, BMI, family history of BC, age at menarche, parity, OC use, smoking, alcohol, education
			postmenopausal women	167		RR 0.86, 95% CI: 0.39, 1.89	
			premenopausal women	180		RR 2.15, 95% CI: 0.95, 4.86	
9 [32]	Gunter, 2015	WHI	postmenopausal women	856	highest vs. lowest quartile	HR 0.82, 95% CI: 0.59, 1.14	age, BMI, ethnicity, alcohol, family history of BC, parity, years of menstruation, age at first birth, HRT use, endogenous estradiol levels, history of BBD, physical activity
6 [29]	Dias, 2016	MDC Cohort	postmenopausal women	446	highest vs. lowest tertile	**OR 0.65, 95% CI: 0.43, 0.99**	age, week of blood sampling, BMI, WHR, HRT use, parity, smoking, alcohol, physical activity, education
***White blood cell count***							
4 [28]	Busch, 2018	WHI	postmenopausal, invasive cancer	4328	dichotomized at 10,000 cells/uL	HR 1.06, 95% CI: 0.87, 1.30	age, race/ethnicity, cohort enrollment, age at menarche, age at menopause, HRT use, breastfeeding, BMI, smoking status, caregiving, negative life events, physical activity, sleep quality
			postmenopausal, in situ cancer	1049		**HR 1.65, 95% CI: 1.17, 2.33**	
6 [29]	Dias, 2016	MDC Cohort	postmenopausal women	446	highest vs. lowest tertile	OR 0.93, 95% CI: 0.67, 1.30	age, week of blood sampling, BMI, WHR, HRT use, parity, smoking, alcohol, physical activity, education
8 [31]	Frydenberg, 2016	Norwegian cohort	all women in study	192	highest vs. lowest tertile	HR 1.04, 95% CI: 0.77. 1.41	age, BMI, number of children, smoking
			postmenopausal women	149		HR 1.03, 95% CI: 0.73, 1.46	
			premenopausal women	43		HR 1.02, 95% CI: 0.54, 1.94	
18 [37]	Wulaningsih, 2015	AMRS cohort	all women in study	2265	dichotomized at 10 10^9^/L	HR 1.07, 95% CI: 0.90, 1.28	age, SES
			postmenopausal women	1960		HR 1.06, 95% CI: 0.88, 1.28	
			premenopausal women	962		HR 1.04, 95% CI: 0.81, 1.32	

Notes: AMRS = Apolipoprotein Mortality Risk Study; BBD = benign breast disease; BC = breast cancer; BMI = body mass index; EPIC = European Prospective Investigation into Cancer and nutrition; HRT = hormone replacement therapy; MDC = Malmö Diet and Cancer; NSHDS = Northern Sweden Health and Disease Study; OC = oral contraceptive; SES = socioeconomic status; WHI = Women’s Health Initiative; WHS = Women’s Health Study; WHR = waist-to-hip ratio; * 95% CI appears as reported in original publication. Unique study # distinguishes the 18 different studies from the 16 publications identified in this review and can be used to cross-reference with Table 1. Citation # corresponds to the reference number of each publication in this review and can be cross-referenced with the Reference list at the end of the paper.

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
