# Peer review of "Inflammatory Biomarkers and Breast Cancer Risk: A Systematic Review of the Evidence and Future Potential for Intervention Research"

_ijerph, 2020, doi:10.3390/ijerph17155445_

Round 1

Reviewer 1 Report

This study conducted a systematic review of prospective studies investigating inflammatory biomarkers in blood in relation to breast cancer risk. My detailed comments are listed below:

Introduction:

  1. Line 65: check the space between “increased” and “cancer risk”
  2. Are all studies shown in Table 2 and 3 included in Table 1? Clarify the differences between Table 1 and Table 2/3.
  3. In Table 2, when there were multiple publications from the same study/cohort (for example, 3 publications from WHI), how did you treat them?
  4. In introduction, I suggest to add a brief explanation of how different biomarkers mentioned in the paper play a role in inflammatory and carcinogenic pathways, especially non-CRP markers. Describe what their levels indicate.

Results:

  1. Page 3, line 104: Why did you decide to include meta-analyses in the review? In line 104, the authors stated the meta-analyses included “mostly” publications from early years. Were there any overlapping studies between those included in previous meta-analyses and the individual studies that authors newly searched?
  2. Page 3, line 108: “ 12 of the 15 new studies”: Which markers did the rest of 3 studies measure?
  3. Page 3, line 109: “eight of the 12 new studies”: How about other 4 studies?
  4. Page 6, first paragraph: Did previous meta-analyses find any heterogeneity across studies (I2)? Any evidence of publication bias?   
  5. Page 6, second paragraph: Is there any information on the break-up of premenopausal vs. postmenopausal women in the three cohort studies mentioned in this paragraph?
  6. Page 6, line 130: Please add reference in the sentence “five studies examined the association of CRP….”
  7. Page 6, line 133, “two studies found a statistically significant….for postmenopausal women only”: In these studies, was there no association with premenopausal women? Is this sentence merely based on statistical significance? Or also based on the point estimate? I wonder if the direction of the association is still positive although non-significant.
  8. Page 6, line 140: “Nelson et al….with a BMI<25 kg/m2” : in the same study, was there no association in postmenopausal women with BMI>25? In Dossus et al, was there no association in women with BMI<25? Please clarify.
  9. Page 6, line 144, “Dossus et al.....waist circumference of >88”: is it the only study stratify by waist circumference?
  10. Page 12, 3rd paragraph: Leptin and adiponectin may also non-inflammatory effects. Please clarify what these markers may indicate.
  11. Results: was there any difference in study results by covariate adjustment? Any difference in assay method? Reliability of markers?

Discussion

  1. Page 13, line 226: “showed a consistent association” seems to be contradictory to results, page 6, line 130-131 “five studies examined the association of CRP…..but inconsistent findings.” Please clarify.
  2. Page 13, line 228, “other inflammatory biomarkers may be more specific to breast-related changes.” Please clarify why. They are all blood markers, not breast tissue markers.
  3. Page 13, line 246-248: Does the results by age support this hypothesis (window of susceptibility)?
  4. I suggest to provide some explanations for the summary findings in results (e.g., explanations for any inconsistency in study findings).
  5. Is there any reason why CRP is more extensively studied in previous studies? Is it easier to measure than other markers? I suggest to add some explanations in the text.

Reviewer 2 Report

The manuscript submitted by Khem et al. is a very good updated review on the advances in recent years in the description of inflammatory biomarkers associated with breast cancer risk. Likewise, it highlights the importance of undertaking new studies to deepen the subject.
It is very well written, with a clear wording and in a concrete way the main findings are described so far, highlighting the importance of these studies to support the development and evaluation of mechanistically driven interventions to reduce inflammation and prevent breast cancer.

Reviewer 3 Report

Discussion is very simple, not a lot of emphasis on the results from the metanalysis and the implications for those.

-Talk about reverse causality and the effect it may have had on the results seen throughout the studies. Follow-up time and mean age at baseline are important elements in these kinds of studies, would like to see how they were different between studies. 

-I would like to see more analysis of the possible different mechanisms between premenopausal and postmenopausal breast cancer in relation to inflammation.

-Confounders like medications (NSAID) are important, did any study took them into account?

Reviewer 4 Report

The authors present a thorough and thoughtful systematic review of the literature on inflammatory biomarkers and risk of breast cancer. The literature is currently limited in terms of the lack of breast tissue-specific inflammatory markers, the varying units of comparison, and sometimes conflicting results across studies. However, there is some indication that elevated CRP, the most commonly measured systemic inflammation biomarker across the reviewed studies, is associated with increased risk of breast cancer and that this risk may be greater among postmenopausal women.

The authors briefly review prior evidence (or rather, lack thereof) regarding other medical conditions that have been associated with both chronic inflammation and breast cancer risk, as well as animal model and human breast tissue morphology studies. They conclude with the suggestion that, “If a strong and independent link is established between inflammation and breast cancer risk, it will be important to conduct mechanistic studies to understand inflammatory pathways and demonstrate how breast tissue responds to chronic inflammation, such as through altered metabolic activity.” (lines 315-318) Based on the results of the review, it seems as if establishing such a link via currently available epidemiologic studies will be challenging. Could existing mechanistic evidence regarding the role of inflammation in breast cancer progression serve as a better starting point for either evaluating the potential role of chronic inflammation in breast cancer risk, and/or identifying better inflammatory markers for prospective evaluation in epidemiologic studies? What do the authors think is the most expedient path forward?

Round 2

Reviewer 1 Report

Authors have adequately addressed the reviewer's comments. 

Reviewer 3 Report

No comments